# Sleep Disturbances and Health Consequences Induced by the Specificity of Nurses’ Work

**DOI:** 10.3390/ijerph19169802

**Published:** 2022-08-09

**Authors:** Małgorzata Knap, Dorota Maciąg, Edyta Trzeciak-Bereza, Bartosz Knap, Marcin Czop, Sabina Krupa

**Affiliations:** 1Institute of Health Sciences, Collegium Medicum of the Jan Kochanowski University, 25-369 Kielce, Poland; 2Faculty of Health Sciences, University of Business and Enterprise, 27-400 Ostrowiec Świętokrzyski, Poland; 3Department of Experimental and Clinical Pharmacology, Faculty of Medicine, Doctoral School, Medical University of Lublin, 20-090 Lublin, Poland; 4Clinical Genetics Department, Medical University of Lublin, 20-080 Lublin, Poland; 5Institute of Health Sciences, College of Medical Sciences, University of Rzeszow, St. Warzywna 1A, 35-310 Rzeszow, Poland

**Keywords:** sleep, nurses, health consequences, work specification

## Abstract

Introduction: Nursing staff working in a shift or night system are exposed to sleep disorders, which has a direct impact on the emergence of dangerous health consequences for them. Melatonin secretion is abnormal at night and the circadian rhythm is disturbed. The aim of the study was to assess the occurrence of sleep disorders and their consequences for the body in a group of representative nursing staff working in a shift and night system. Participants: The study was conducted among 126 nurses who are generally healthy, employed in health care facilities in the Małopolskie voivodship. Methods: The Athens Insomnia Scale consisting of 8 test items was used to obtain research material: falling asleep, waking up at night, waking up in the morning, total sleep time, sleep quality, well-being the next day, mental and physical fitness the next day, and sleepiness during the next day. As well as an original questionnaire. Results: The research showed significant negative consequences of shift work on the health of health-care workers. The subjects noticed symptoms related to the nervous system, such as increased nervous tension 53%, lack of patience in 62% of all respondents. As many as 85% pointed to the negative impact of shift work on their family life, 82% of all respondents on social life and 56% of all respondents on sex life. The other variables were not confirmed. Conclusions: Symptoms of insomnia are common among night-work nurses.

## 1. Introduction

Employees of medical entities provide health services on the basis of legal regulations, so their working time is regulated in Poland by the Act of 15 April 2011 on medical activity (Journal of Laws No. 112, item 654, as amended). In Poland, we have various forms of employment, but most often the nursing staff is employed in a two-shift system (where a day is divided into two 12-h shifts), which is associated with workload during the night [1]. Scientific literature indicates that disturbances of the circadian rhythm in melatonin secretion, caused by a work shift, may affect a number of health problems [2]. The most important factor for human functioning is the effect of artificial lighting, which suppresses the secretion of melatonin and disrupts the basic rhythm of wakefulness–sleep, and the astronomical day–night cycle [3]. After many years of working in a shift system, there are numerous health consequences in the body, occurring at the level of many systems (nervous, cardiovascular, digestive, endocrine, thermoregulatory, immune), as well as sleep disorders [4]. Even a disease entity has been identified, which is called shift work intolerance syndrome in the International Classification of Sleep Disorders. Symptoms of this disease, such as chronic fatigue, sleep disorders, digestive system diseases, psychoneurotic ailments, may appear after a few months or several years of shift work [5]. In 2007, the International Agency for Research on Cancer (IARC) concluded that the circadian rhythm disturbances associated with shift work are a potentially carcinogenic factor [6]. A significant correlation was found between the incidence of breast cancer in women and the number of night shifts per month. The risk of breast cancer incidence was considered significant when the woman worked more than two nights per month and the length of shift work was long [7,8]. An epidemiological study from 2015 published by doctors from the American Thoracic Society confirmed that sleep duration less than 6 or more than 9 h a day and poor sleep quality lead to the occurrence of many diseases [9]. According to experts from the National Sleep Foundation, the recommended length of sleep depends on age and for people between 18 and 65 years of age it should be between 7 and 9 h a day [10]. Nursing staff working at night are prone to sleep and wake disorders. There are possibilities to prevent the health effects of shift work, but it is impossible to eliminate the negative effects of night work. People responsible for the organization of shift work should take steps to improve working conditions and plan the time of professional activity so that it is adapted to the psychophysical abilities of the employee [11]. The Directive of the European Parliament and of the Council on certain aspects of the organization of working time, states that the situation of night and shift workers requires that the level of protection of their safety and health should be adapted to the nature of the work [12]. When taking action to prevent the effects of shift work, you should start with primary prevention, which includes health promotion, compliance with the provisions of the Labor Code, and compliance with the recommendations of the World Health Organization and EU Directives relating to shift workers and night workers. Work ergonomics and health education of employers and employees regarding potential health hazards at work with excessive night shifts are still underestimated and requiring attention in areas of health care institutions. Proper medical care is also important for maintaining the health of the employee [11].

### Aim of the Study

Assessment of the occurrence of sleep disorders and their consequences for the body in the group of representatives of nursing staff working in a shift and night system.

## 2. Materials and Methods

### 2.1. Study Design

A prospective repeated-measure descriptive study was conducted. An original questionnaire and AIS were used to obtain the research material. Participation in the study was voluntary, data for the study were obtained anonymously. The respondents provided the required answers after prior consent to participate in the study, and they also received instructions on how to use the research tools. The next stage of the research was subjecting the collected research material to statistical analysis in order to formulate conclusions from the research.

### 2.2. Sample and the Tool

The survey was conducted among 126 nurses employed in hospitals in Małopolska, in accordance with the guidelines contained in the Helsinki Declaration. The calculated sample size for this study was 104 participants. Before starting the study, each person was informed that participation in it is voluntary and anonymous, and the results will be used only for research purposes. The paper uses the method of a diagnostic survey with the use of a survey research technique. To obtain the research material, a proprietary questionnaire containing 41 questions and the Athens Insomnia Scale (AIS) were used. The AIS tool was validated for Polish conditions in 2011 [13]. The AIS is an eight-point scale that allows you to quantify the symptoms of insomnia. It is the most commonly used scale, in the diagnosis of insomnia, as well as in the study of the effectiveness of the treatment of insomnia. The original validation studies demonstrated high reliability and validity of this tool. The total score on a scale of 6 or more points was considered a value that allowed to conclude with a high probability about the presence of insomnia in the subject. The AIS scale is a self-descriptive tool, it consists of eight statements, each of them relating to different symptoms of insomnia. The first five items relate to sleep symptoms: quality and duration. The symptom should be marked if it has occurred at least three times a week for at least one month. The next three items concern functioning during the day: mental and physical fitness, well-being and sleepiness. Each item is assessed on a scale of 0–3 points, where 0 means no symptom, and 3 means significant severity. The subjects could get a score in the range of 0–24 points. The AIS is the first Polish validated tool for the assessment of symptoms related to insomnia. The research confirmed the good psychometric properties of the scale. A total score in AIS of 8 points and more was considered a value that allowed concluding with a high probability of the presence of non-organic insomnia according to the ICD-10 criteria. The brevity, reliability and accuracy of AIS makes this tool useful in clinical practice and research [13]. The results of the AIS scale made it possible to divide the study group into levels of insomnia and assign them responses in accordance with the results of the self-authored questionnaire. The survey questionnaire included the use of stimulants (strong coffee, strong tea, high-energy drinks and smoking nicotine) and the frequency of using drugs to help you fall asleep. The questionnaire did not include questions about the use of other medications by the respondents.

### 2.3. Statistical Analysis

The results obtained on the nominal and ordinal scales were presented as the number (n) and percentage (%). In the statistical analysis, the χ^2^ test of independence was used to assess the relationship between the studied variables on the nominal and ordinal scales. Additionally, univariate and multivariate logistic regression analysis was carried out in order to accurately assess the influence of the studied factors on insomnia. Statistical analysis was performed using the Statistica v.13.0 program (StatSoft, Kraków, Poland). The results at *p* < 0.05 were considered statistically significant.

### 2.4. Participants

126 people took part in the study, and because nursing is the domain of women, the study group consisted of 123 women (98%) of the total number of respondents and 3 men, which constituted 2% of the total number of respondents. In the group of respondents aged 20–30, there were 22% of the total number of respondents, 20% aged 31–40, and 10% aged 51–60. The group of nurses aged 41–50 years was the most numerous among the respondents, constituting 48% of the total (61 people). Work experience in a shift system is essential for the research. In the group of respondents, 31% worked shifts for less than 10 years, 23.8% of all respondents worked 11–20 years, 8.7% worked for 31–40 years, and 1% of the respondents worked shifts for over 40 years. The largest group of people, were the 45 (35.7%) people working in shifts for 21–30 years. For the reliability of the research on sleep disorders in nurses, the shift workload during the night was verified. Among the respondents, 3–4 night shifts per month were declared by 15% of the respondents, 21% of the respondents worked at night 7–8 times a month, more than 8 night shifts per month were declared by 10% of the respondents. The most numerous group of 68 people (54%) were people working on night shifts 5–6 times a month. The data is presented in Appendix A.

### 2.5. Inclusion and Exclusion Criteria

***Inclusion criteria*:** adults, nurses, people working for a minimum of 3 months in the profession.

***Exclusion criteria*:** students, other health care professionals other than nurses, people with less than 3 months of work experience as a nurse.

### 2.6. Ethical Considerations

The study was conducted according to the guidelines of the Declaration of Helsinki and approved by the Bioethics Committee of the University of Rzeszów (KBE No. 9 May 2020).

## 3. Results

The study analyzed the age, length of work in the profession and the number of shifts that may have an impact on sleep disorders and insomnia.

Based on the analyzed data, there is a statistically significant correlation between the risk of insomnia and clinical insomnia with regard to age, work experience and shift work, while no correlation is found for the number of night shifts. Additionally, the study analyzed the atmosphere at the workplace, the assessment of well-being and the phenomenon of fatigue at work. The results are presented in Table 1.

Most of the respondents (41.1%) assessed the atmosphere at the workplace as pleasant and friendly. Similarly, they assessed their well-being as good (40.35%). The respondents replied (18.18%) that they are sleepless at work several times a week and it is this group of respondents that manifests clinical insomnia in over 36%. Based on the analyzed data, a statistically significant correlation is found between the risk of insomnia and clinical insomnia in relation to the atmosphere at the workplace and episodes of lack of sleep at work. However, no statistically significant relationships were found for the assessment of well-being at work. There was a much higher risk of insomnia and clinical insomnia in the group of respondents who rated the work atmosphere as tense and nervous compared to the respondents who assessed the atmosphere as pleasant and friendly (OR = 3.59; *p* = 0.0074). A much lower risk of insomnia and clinical insomnia was found in the group of respondents who do not feel sleepy at work (OR = 0.22; *p* = 0.0013) and feel sleepy at work occasionally (OR = 0.07; *p* = 0.0294) compared to respondents who feel sleepy at work several times a month. Additionally, analyzing the risk of clinical insomnia only, it was shown that it is lower in the group of respondents who feel asleep at work occasionally (OR = 0.09; *p* = 0.0005) compared to respondents who feel sleepy at work several times a month (Table 2). The occurrence of episodes of loss of self-control and the occurrence of episodes of outbursts/nervous tension at work among the surveyed nurses were subjected to statistical analysis (Table 3).

The respondents assessed that quite often (55.56%) they lose control over themselves, which results from fatigue after work. This group was the most numerous group that had symptoms of clinical insomnia. More than 17% of people with clinical insomnia do not have such problems. More than half of the respondents (54.29%) represent the risk of developing insomnia. The group of 18.18% with clinical insomnia believes that they never have any emotional problems at work. Based on the analyzed data, a statistically significant correlation is found between the risk of insomnia and clinical insomnia in relation to the loss of self-control due to fatigue. Adverse behavior during night shifts as adverse effects of shift work in the studied group included: mood change, frustration, depression, hostility, helplessness, irritation, irritation, nervousness, and a sense of anxiety. The occurrence of these episodes among the respondents and the risk of sleep disorders in nurses were statistically analyzed (Table 4).

Thirty percent of people with clinical insomnia believe that they experience frustration after a night shift. Such symptoms are represented by 60% of respondents who are at risk of developing insomnia. A group of 2 people (10%) believe that they experience frustration, but their results do not indicate sleep problems. The feeling of depression was characteristic in the group of people at risk of insomnia in 16.67%. More than half of the respondents (66.67%) feel depressed and they show clinical insomnia, in line with the results. Forty percent of the respondents feel hostile to the environment, who are susceptible to the risk of insomnia and people with clinical insomnia. Helplessness is a feeling that is characteristic of half of the respondents with clinical sleepiness. In turn, annoyance occurs in 50% of respondents who are at risk of developing insomnia. In this group of respondents, irritation occurs in almost half of the respondents (48.15%). The group at risk of developing insomnia exhibits 50% nervousness. More than half of the respondents with clinical insomnia feel a sense of anxiety. A higher risk of insomnia and clinical insomnia was shown in the group of respondents who experienced mood changes such as frustration (OR = 5.03; *p* = 0.0365), helplessness (OR = 2.64; *p* = 0.0194) and irritation (OR = 3.29; *p* = 0.0405) compared to respondents who did not have such mood changes. Based on the analyzed data, there is a statistically significant correlation between the risk of insomnia and clinical insomnia with respect to such disorders as: frustration, helplessness, irritability, irritation, nervousness and a sense of anxiety. There are no statistically significant correlations for such emotional problems as depression, hostility and nervousness. Nurses working on night shifts experience various indispositions such as heartburn, flatulence and diarrhea. The occurrence of digestive tract disorders in the studied nurses was analyzed in relation to the risk of insomnia and clinical insomnia (Table 5).

The respondents were also asked about ailments related to the digestive system. Almost half (45.71%) of the respondents have a problem with constipation. This group has symptoms of risk of insomnia. Forty-five point 5 percent of nausea occurs in people diagnosed with clinical insomnia. A total of 48.57% of people at risk of insomnia reported heartburn. Half of the respondents have problems with diarrhea and experience clinical insomnia. Moreover, half of the respondents have a problem with flatulence and this is a group at risk of developing insomnia. A total of 53.57% of this group experience epigastric pain. Half of the respondents who are at risk of developing insomnia believe that they suffer from digestive problems. A quarter of respondents who have clinical insomnia believe that they do not have problems related to the digestive system. Based on the analyzed data, there is a statistically significant relationship between the risk of insomnia and clinical insomnia with respect to gastrointestinal disorders. There was a higher risk of insomnia and clinical insomnia in the group of respondents who suffer from heartburn (OR = 2.88; *p* = 0.0335) and flatulence (OR = 2.44; *p* = 0.0281) compared to the respondents, who do not have such symptoms. Additionally, analyzing the risk of clinical insomnia only, it was shown that it is higher in the group of respondents who suffer from heartburn (OR = 4.25; *p* = 0.0274) and flatulence (OR = 3.11; *p* = 0.0281) compared to respondents who do not have such symptoms. The incidence of cardiovascular disorders in the studied nurses was analyzed in relation to the risk of insomnia and clinical insomnia (Table 6).

Almost half of the respondents (48.94%) at risk of insomnia believe that they have problems with the cardiovascular system, such as palpitations. There was a higher risk of insomnia and clinical insomnia in the group of respondents who suffer from heart palpitations (OR = 2.65; *p* = 0.0544) compared to respondents who do not have such symptoms. Additionally, analyzing the risk of clinical insomnia only, it was shown that it is higher in the group of respondents who suffer from heart palpitations (OR = 3.00; *p* = 0.0366) compared to respondents who do not have such symptoms. An irregular heartbeat occurs in half of the respondents who are at risk of insomnia. A total of 31.25% of people with clinical insomnia have symptoms related to irregular heartbeat. Increased CTK values are characteristic in 34.48% in people at risk of insomnia, and in people with clinical insomnia, it occurs in 31.03%. Accelerated heart rate is characteristic of people at risk of insomnia in 39.13%, and 32.61% have tachycardia in the clinical insomnia group. Almost half (47.37%) of the respondents believe that they have no problems with the cardiovascular system. Based on the analyzed data, there is no statistically significant correlation between the risk of insomnia and clinical insomnia in relation to cardiovascular disorders. The occurrence of changes in behavior and the manner of reacting in the surveyed nurses with regard to the risk of sleep disorders was analyzed (Table 7).

Half of the respondents at risk of insomnia experience mood swings after night shift. More than half (55.88%) of people with clinical insomnia experience constant fatigue. 53.85% of respondents with clinical insomnia noticed the presence of aggression. A higher risk of insomnia and clinical insomnia was found in the group of respondents who, as a result of working night shifts, experience changes in mood (OR = 3.42; *p* = 0.0023), and experience constant fatigue (OR = 11.26; *p* < 0.00001) compared to respondents who do not have such changes in mood and feelings. Less than half of the respondents (47.62%) notice the presence of criticism. More than half of the respondents (52.38%) in the group of people with clinical insomnia noticed exaggerated reactions in their way of reacting. Anxiety occurs in 61.54% of patients with clinical insomnia, and in 15.38% of patients at risk of insomnia. Based on the analyzed data, a statistically significant correlation is found between the risk of insomnia and clinical insomnia with regard to mood changes, fatigue, changes in behavior such as outbreaks of aggression, the occurrence of criticism and the risk of anxiety or anxiety. Additionally, analyzing the risk of occurrence of clinical insomnia only, it was shown that it is higher in the group of respondents who experience mood changes as a result of working night shifts (OR = 5.08; *p* = 0.0055), experience constant fatigue (OR = 40, 11; *p* <0.00001), show aggression (OR = 6.33; *p* = 0.0292), show criticism (OR = 6.85; *p* = 0.0074), show an overreaction (OR = 5, 82; *p* = 0.0070) and experience anxiety (OR = 4.93; *p* = 0.0292) compared to respondents who do not have such changes in mood and feelings (Table 7).

The respondents were asked about how they spend their free time, which also has a huge impact on sleep. A total of 72% of respondents spend their free time with their family, 46% of people spend time in the garden, 45% of respondents watch TV or read books in their free time. Just over 40% of the respondents spend their free time walking, and 21% of the respondents ride a bicycle in their free time. A total of 23% of the respondents spend their free time in the cinema, 15% do gymnastics in their free time, 11% swim, and 6% of the respondents spend their time alone. Only 2% of the respondents do not take up any activities in their spare time.

According to the results, in the group of people at risk of insomnia, more than half of them do not spend their free time in gymnastics. A total of 63.16% of people who do not have problems with insomnia use gymnastic exercises in their spare time. People with clinical insomnia use 15.79% of their free time for exercises. Free time reading is characteristic mainly for people who do not have problems with insomnia. A lower risk of insomnia and clinical insomnia was shown in the group of respondents who spend their free time doing gymnastics (OR = 0.21; *p* = 0.0026) and reading (OR = 0.42; *p* = 0.0247) in compared to respondents who do not perform such activities. More than 50% (57.97%) of people at risk of insomnia do not spend time reading books. The use of various forms of spending time in the surveyed nurses was statistically analyzed in relation to the risk of insomnia and clinical insomnia (Table 8).

Privacy and family life are an essential part of life. The family is support for a man, it provides help in difficult times. Based on the analyzed data, a statistically significant correlation is found between the risk of insomnia and clinical insomnia with regard to two forms of spending time (Table 9).

The most numerous group was the people who considered that shift work has a moderately negative impact on family life (52%), slightly less, 33% of respondents believed that shift work has a definitely negative impact on family life. In the studied group, 13% of respondents considered that shift work did not affect family life, and only 2% of respondents considered that shift work had a moderately positive effect on family life. The respondents assessed the influence of shift work on social life in a similar way. The most numerous group was the people who considered that shift work has a moderately negative impact on social life (44%), slightly less—38% of respondents believed that shift work has a definitely negative impact on social life. Only 13% of respondents believed that shift work did not affect social life, and only 2% of respondents considered that it had a moderately positive effect on social life. Working at night may predispose you to consume high-energy drinks. The study analyzed the consumption of high-energy drinks, which are stimulants that affect health, CNS and behavior. However, no statistically significant correlation with the problem of sleep disorders was found. There was a higher risk of insomnia and clinical insomnia in the group of respondents who noticed a relationship between the amount of coffee/strong tea drunk with night changes (OR = 2.44; *p* = 0.0281) compared to respondents who did not notice such a relationship. Working at night disturbs the rhythm of sleep, leading to the formation of disorders, to facilitate falling asleep, people use sleeping pills, unfortunately even the latest generation preparations can lead to unfavorable consequences. Based on the analyzed data, there is a statistically significant correlation between the risk of insomnia and clinical insomnia with regard to the use of sleeping pills among the surveyed nurses (Table 9).

The negative effects of night work may damage your health. Lack of sleep and constant fatigue can lead to conflict situations. The study assessed the nurses’ tendency to conflict behavior and their own appearance (Table 10).

Forty percent of both people with clinical insomnia and people at risk of insomnia believe that they are conflicting people. Almost half of the respondents (47.06%) at risk of insomnia care about their appearance, and 100% of people diagnosed with clinical insomnia do not care about their appearance. A lower risk of insomnia and clinical insomnia was shown in the group of respondents who sometimes forget about their appearance (OR = 0.02; *p* < 0.00001) compared to respondents who always pay attention to their appearance. On the other hand, analyzing the risk of clinical insomnia only, it was shown that it is higher in the group of respondents who sometimes forget about their appearance (OR = 4.31; *p* = 0.0081) compared to respondents who always pay attention to their own appearance. On the basis of the analyzed data, a statistically significant correlation was found between the risk of insomnia and clinical insomnia in relation to two elements, i.e., own appearance and increased body weight (Table 10).

More than 41% of respondents at risk of insomnia have problems related to menstrual disorders. Almost half of people at risk of insomnia have painful (48.57%) and heavy periods (51.52%). There was a higher risk of insomnia and clinical insomnia in the group of female respondents who have heavy menstruation (OR = 6.07; *p* = 0.0051) compared to respondents who do not show such symptoms. Additionally, analyzing the risk of clinical insomnia only, it was shown that it is higher in the group of female respondents who have abundant menstruation (OR = 9.82; *p* = 0.0013) and have painful menstruation (OR = 3.88; *p* = 0.0211) compared to respondents who did not show such symptoms. Less than 50% (49.48%) of people have irregular periods in the group of people at risk of insomnia. In almost half of the people at risk of insomnia (46.79%), spontaneous miscarriage occurred during the years of shift work.

Among the respondents, a statistically significant relationship is found for disorders such as excessive profusion (*p* = 0.0006) and menstrual pain (*p* = 0.0357). People who did not have these disorders constituted a more numerous group. Additionally, analyzing the risk of clinical insomnia only, it was shown that it is lower in the group of female respondents who have normal menstruation (OR = 0.23; *p* = 0.0131) compared to the respondents who have any menstrual irregularities (Table 11).

Moreover, the respondents indicated other disorders of their own body functioning.

It is found among the respondents that working in a shift system causes a statistically significant dependence and affects sleep disorders, for disorders such as problems with concentration and decision-making, problems with memory, increased susceptibility to accidents or making mistakes. It is also found among the respondents that there are visual disturbances and the need to wear glasses, which causes a statistically significant correlation and affects sleep disorders. A statistically significant relationship is found among the respondents when analyzing migraine headaches and sleep disturbances in shift work. A higher risk of insomnia and clinical insomnia was shown in the group of respondents who show a negative attitude (OR = 1.55; *p* = 0.0346), have memory problems (OR = 2.04; *p* = 0.0070), have increased susceptibility to accidents (OR = 1.61; *p* = 0.0287), make mistakes more often (OR = 1.70; *p* = 0.0192), are under the care of an ophthalmologist (OR = 1.53; *p* = 0.0314) and suffer from headache and migraine (OR = 1.63; *p* = 0.0126) compared to respondents who do not show such symptoms. Additionally, analyzing the risk of clinical insomnia only, it was shown that it is greater in the group of respondents who show a negative attitude (OR = 3.52; *p* = 0.0158) and have memory problems (OR = 8.08; *p* = 0.0006), make mistakes more often (OR = 3.47; *p* = 0.0232) and are under the care of an ophthalmologist (OR = 4.64; *p* = 0.0040) compared to respondents who do not show such symptoms. Detailed results are presented in Table 12.

### Multivariate Logistic Regression

A univariate logistic regression was carried out in order to define the exact influence of the studied factors on the phenomenon of insomnia among the subjects. In the next step, a multivariate logistic regression was performed for those factors, the result of which was statistically significant in the univariate analysis.

There was a higher risk of insomnia and clinical insomnia in the group of female respondents who notice constant fatigue (OR = 10.22; *p* = 0.0115) and have heavy menstruation (OR = 5.40; *p* = 0.0460). Additionally, it has been shown that the risk is reduced in female respondents who occasionally feel sleepy at work (OR = 0.12; *p* = 0.0015) and read in their spare time (OR = 0.13; *p* = 0.0008). Additionally, analyzing the risk of clinical insomnia only, it was shown that it is greater in the group of female respondents who notice constant fatigue (OR = 9.81; *p* = 0.0164), have heavy menstruation (OR = 20.37; *p* = 0.0207), are under the care of an ophthalmologist (OR = 19.72; *p* = 0.0125) and have memory problems due to night work (OR = 21.26; *p* = 0.0170) (Table 13).

A higher risk of insomnia and clinical insomnia was shown in the group of respondents who notice constant fatigue (OR = 3.40; *p* = 0.0442), are under the care of an ophthalmologist (OR = 3.39; *p* = 0.0255), memory problems as a result of night work (OR = 4.40; *p* = 0.0391). Additionally, it has been shown that this risk is reduced in respondents who occasionally feel sleepy at work (OR = 0.20; *p* = 0.0071), read in their free time (OR = 0.19; *p* 0.0214) and practice leisure time gymnastics (OR = 0.20; *p* = 0.0109). Additionally, analyzing the risk of clinical insomnia only, it was shown that it is greater in the group of respondents who are under the care of an ophthalmologist (OR = 54.29; *p* = 0.0010) and have problems with memory as a result of working on night work (OR = 38, 88; *p* = 0.0027), have heart palpitations (OR = 14.25; *p* = 0.0103). Additionally, it has been shown that this risk is reduced in respondents who occasionally feel sleepy at work (OR = 0.06; *p* = 0.0080) (Table 14).

## 4. Discussion

Sleep is a very important element of human physiology; it is an irreplaceable form of rest that allows undisturbed functioning of the body. In recent years, many studies have been carried out on the relationship between night work and the occurrence of ailments and diseases in shift workers. The authors of the publication confirm that sleep is an important determinant of the level of health and an element of human lifestyle, along with other health behaviors, such as eating and physical activity. For many nurses, shift work is part of the profession. This solution of work organization was introduced in order to guarantee patients 24/7 care [14]. Reversing the physiological order of the day by working at night and sleeping during the day may be the cause of many ailments among nursing staff, as reported by Serzysko et al. In the course of the study, they obtained results in which 62.5% of respondents suffered from sleep disorders. The most frequently reported complaint was the problem with falling asleep—19.3%, insomnia occurred in 23%, and 7.75% of respondents woke up during sleep. The studies conducted by Szymańska—Czechór M. and Kędra E. obtained similar results, 66.3% of respondents complained of sleep disorders, 43% woke up at night, and 1% of insomnia [15,16]. In the presented analysis of the results of our own research, the presence of insomnia was confirmed in 46% of the respondents. Szymńska, Czechór and Kędra revealed circulatory system problems in nurses in the night system. These were arrhythmias and increased blood pressure. The obtained results indicate the need to extend research in this area [16].

In the studies conducted by Serzysko and associates, attention was drawn to ailments from the nervous system, such as: irritability occurring in 38% of the respondents, an increase in emotional tension occurring in 36% of the respondents, while 18% drew attention to the lack of patience, fatigue occurred in 13% of the respondents [15]. In the studies by M. Szymańska-Czechór and E. Kędra, changes in the nervous system were also noticed. Irritation was experienced by 73.5% of respondents, constant fatigue was experienced by 53%, and 19.3% felt apathy [16]. In our own research, 53% of people noticed an increase in nervous tension, 62% lack of patience, while fatigue occurred in 37% of the respondents, irritation concerned 43% of the respondents, irritation concerned 21% and constant fatigue affected 27% of the respondents, and drowsiness occurred in 61% of the respondents. An important element of the proper functioning is family life, intimate life and satisfaction with sex life. In the studies by Zużewicz et al., a decrease in libido and sexual performance as an effect of many years of working at night was found, but this conclusion was not confirmed by any detailed analysis [17]. In the studies by Kasperczyk and Jośko, 76.2% of respondents indicated a negative impact of shift work on family life, 17.2% had no impact, and 6.6% of respondents indicated a moderately positive impact on family life [18]. Similarly, the respondents answered the question about the impact of shift work on social life: 74.2% of the respondents considered that it had a negative impact, 20.5% found that it had no influence, and 5.3% found that it had a moderately positive effect. In a study by Chang et al., Nurses have to change their daily activities frequently, which affects their biological circadian rhythm and can cause sleep disturbance and fatigue [19]. In the presented analysis of the obtained research results, it was noticed among 52% of respondents that shift work has a moderately negative impact on family life. Lower results among the respondents were obtained by asking about the impact of shift work on social life, 38% of respondents considered that shift work has a definitely negative impact on social life, while 44% of respondents considered that shift work has a moderately negative impact on life. A total of 56% of respondents complained about decreased sexual activity. In the studies by Serzysko et al., The respondents indicated complaints from the digestive system: flatulence affected 39%, abdominal pain occurred in 17%, 12.5% suffered from constipation, burning sensation in the esophagus affected 11%, diarrhea occurred in 9%, and nausea in 4.5% of respondents [15]. In the course of studies by Szymańska-Czechór M. and Kędra E., 49.4% of respondents noticed changes in the functioning of the digestive system, 50.6% indicated disturbances in intestinal peristalsis, and 15.7% indicated epigastric pain [16]. The results of our own research were similar. In 44% of the respondents the symptoms were flatulence, 28% suffered from constipation, also 28% had heartburn, 9% had nausea, and 8% complained of diarrhea, 22% complained of abdominal pain; at the same time, 30% of all respondents did not report any problems with the digestive system. The results of epidemiological studies indicate that reduced sleep duration may be a risk factor for the development of cardiovascular diseases. This thesis is confirmed by the Polish research by Pol–Monica Bis, which reports that people with insomnia are characterized by a more frequent occurrence of cardiovascular diseases [20,21]. In Japan, Kubo et al. conducted an interesting experiment with 36 nurses using Doppler ultrasound to assess coronary blood flow. Nurses were examined twice, on the day before the night shift and in the morning after the night shift. The results of the research indicated a deteriorated coronary microcirculation in nurses working on the night shift [22]. In studies conducted by Szymańska-Czechór and Kędra, 36.1% of respondents indicated problems with the circulatory system. Among the complaints, 36.1% indicated arrhythmia and palpitations, and 32.5% reported elevated blood pressure [16]. Our own research obtained similar results concerning the cardiovascular system. Palpitations and increased heart rate were indicated by 37% of people, 23% of respondents indicated increased arterial blood pressure, and 13% experienced an irregular heartbeat. In the group of respondents, only 30% of respondents did not report any problems with the circulatory system. Physical activity can also help to improve sleep. In the article, Okechukwu et al., emphasize that following a moderate-intensity aerobic exercise program can improve both the quality and duration of sleep [23]. In turn, studies by Querstret et al. showed that longer shifts, including nights, and insufficient recovery time between shifts were associated with worse sleep, increased sleepiness and increased fatigue [24]. According to many studies, sleep interventions should be developed that can positively promote the health of nurses and promote effective work performance [25]. In addition, hospital decision makers should consider developing new guidelines to minimize the negative effects of night rotation on the mental health and quality of life of nurses [26].

The obtained research results confirmed a significant impact of shift work on the health of nursing staff. Some problems can be solved through primary prevention, while in the event of serious illness, family or social problems, it would be adequate to temporarily or permanently use an alternative form of work to withdraw the employee from work during night hours, for the time appropriate to his health. As a result of the conducted research, it can be concluded that the number of night shifts in a month (5 and more) may lead to the adaptation of changes in the circadian rhythm, which in our opinion is an interesting observation worth noting. All studied groups showed a lower risk of insomnia compared to the group of nurses working on 3–4 night shifts (a statistically significant result of logistic regression was shown for the group of nurses working with 7–8 night shifts; the remaining results were statistically insignificant but close to the value of 0.05).

## 5. Conclusions

The specificity of shift work has negative health consequences (especially for sleep problems). Most of the subjects suffered from sleep problems, neurological problems, and social problems. All these elements were related to night work. The respondents also complained about problems related to the gastrointestinal tract. Another problem of night nurses is the risk of developing cardiovascular diseases. People who do not have insomnia problems do not have various health and interpersonal problems compared to those who have a risk of insomnia and suffer from clinical insomnia.

## 6. Study Limitations

The study was conducted on a group of nurses working night shifts. The study may have limited the various disease entities that may have affected the study group. In addition, the greater number of night shifts could have had an impact on the answers given. Research should also be carried out on other professional groups and the results should be compared also on the basis of research related to possible accompanying disease entities in this group.

## Figures and Tables

**Table 1 ijerph-19-09802-t001:** Age and seniority in the shift system of the surveyed nurses.

Athens Insomnia Scale (AIS)							*p* #	Insomnia Risk + Clinical Insomnia (vs. Norm)		Clinical Insomnia (vs. Norm)	
	Norm		Insomnia Risk		Clinical Insomnia						
	N	%	N	%	N	%					
Age								OR (95% CI)	*p* *	OR (95% CI)	*p* *
20–30	8	28.57%	13	46.43%	7	25.00%	0.0055	1.00	-	1.00	-
31–40	9	36.00%	14	56.00%	2	8.00%		0.71 (0.22–2.26)	0.5637	0.25 (0.04–1.60)	0.1438
41–50	20	32.79%	30	49.18%	11	18.03%		0.82 (0.31–2.18)	0.6911	0.63 (0.18–2.20)	0.4677
51–60	3	25.00%	1	8.33%	8	66.67%		1.20 (0.26–5.61)	0.8168	3.05 (0.57–16.19)	0.1910
Work experience in the profession of a nurse in years											
0–10	13	33.33%	18	46.15%	8	20.51%	0.0389	1.00	-	1.00	-
11–20	7	30.43%	12	52.17%	4	17.39%		1.14 (0.38–3.47)	0.8136	0.93 (0.21–4.21)	0.9234
21–30	17	32.08%	27	50.94%	9	16.98%		1.06 (0.44–2.56)	0.8988	0.86 (0.26–2.84)	0.8051
31–40	3	27.27%	1	9.09%	7	63.64%		1.33 (0.30–5.88)	0.7041	3.79 (0.76–19.05)	0.1056
Work experience in a shift system in years											
0–10	11	28.21%	20	51.28%	8	20.51%	0.0469	1.00	-	1.00	-
11–20	11	36.67%	13	43.33%	6	20.00%		0.68 (0.25–1.88)	0.4557	0.75 (0.20–2.89)	0.6759
21–30	14	31.11%	24	53.33%	7	15.56%		0.87 (0.34–2.23)	0.7715	0.69 (0.19–2.49)	0.5678
31–40	4	33.33%	1	8.33%	7	58.33%		1.05 (0.23–4.69)	0.9515	2.41 (0.52–11.10)	0.2604
Number of night shifts per month											
3–4	2	10.53%	11	57.89%	6	31.58%	0.3529	1.00	-	1.00	-
5–6	22	32.35%	32	47.06%	14	20.59%		0.25 (0.05–1.16)	0.0763	0.21 (0.04–1.20)	0.0798
7–8	11	40.74%	10	37.04%	6	22.22%		0.17 (0.03–0.90)	0.0365	0.18 (0.03–1.20)	0.0762
More than 8	5	41.67%	5	41.67%	2	16.67%		0.17 (0.03–1.06)	0.0575	0.13 (0.01–1.32)	0.0848

N—number of respondents; %—percent; *p* #—statistical significance of the Chi square test; *p* *—statistical significance of univariate logistic regression; OR—odds ratio; 95% CI—95% confidence interval.

**Table 2 ijerph-19-09802-t002:** Atmosphere in the workplace, assessment of well-being and the phenomenon of fatigue at work.

Athens Insomnia Scale (AIS)							*p* #	Insomnia Risk + Clinical Insomnia (vs. Norm)		Clinical Insomnia (vs. Norm)	
	Norm		Insomnia Risk		Clinical Insomnia						
	N	%	N	%	N	%					
How do you rate the atmosphere in the workplace?								OR (95% CI)	*p* *	OR (95% CI)	*p* *
nice and friendly	30	41.10%	26	35.62%	17	23.29%	0.0170	1.00	-	1.00	-
indifferent and cool	3	33.33%	6	66.67%	0	0.00%		1.40 (0.32–6.02)	0.6552	N/A	N/A
tense and nervous	7	16.28%	25	58.14%	11	25.58%		3.59 (1.41–9.13)	0.0074	2.77 (0.91–8.49)	0.0740
enemy and full of aggression	0	0.00%	1	100.00%	0	0.00%		N/A	N/A	N/A	N/A
How do you rate your well-being at work?											
good	23	40.35%	20	35.09%	14	24.56%	0.0910	1.00	-	1.00	-
bad	2	13.33%	8	53.33%	5	33.33%		4.40 (0.91–21.35)	0.0662	4.11 (0.70–24.10)	0.1176
normal	15	27.78%	30	55.56%	9	16.67%		1.76 (0.79–3.90)	0.1648	0.99 (0.34–2.85)	0.9788
Do you ever feel sleepy at work?											
several times a month	9	17.65%	26	50.98%	16	31.37%	0.009	1.00	-	1.00	-
a few times a week	4	18.18%	10	45.45%	8	36.36%		0.96 (0.26–3.54)	0.9563	1.13 (0.26–4.80)	0.8737
occasionally	24	48.98%	21	42.86%	4	8.16%		0.22 (0.09–0.56)	0.0013	0.09 (0.03–0.36)	0.0005
never	3	75.00%	1	25.00%	0	0.00%		0.07 (0.01–0.77)	0.0294	N/A	N/A

N—number of respondents; %—percent; *p* #—statistical significance of the Chi square test; *p* *—statistical significance of univariate logistic regression; OR—odds ratio; 95% CI—95% confidence interval; N/A—Not Applicable.

**Table 3 ijerph-19-09802-t003:** The state of losing control over oneself and the occurrence of episodes of outbursts/nervous tension at work.

Athens Insomnia Scale (AIS)							*p* #	Insomnia Risk + Clinical Insomnia (vs. Norm)		Clinical Insomnia (vs. Norm)	
	Norm		Insomnia Risk		Clinical Insomnia						
	N	%	N	%	N	%					
Do you ever lose control of yourself as a result of tiredness after work?								OR (95% CI)	*p* *	OR (95% CI)	*p* *
never	19	45.24%	18	42.86%	5	11.90%	.	1.00	-	1.00	-
rarely	21	31.82%	32	48.48%	13	19.70%		1.77 (0.80–3.93)	0.1610	2.35 (0.71–7.84)	0.1637
often	0	0.00%	8	44.44%	10	55.56%		N/A	N/A	N/A	N/A
Nervous/emotional “outbursts” at work?											
Yes, often	1	16.67%	1	16.67%	4	66.67%	0.1310	1.00	-	1.00	-
Occasionally	7	20.00%	19	54.29%	9	25.71%		0.80 (0.08–7.99)	0.8493	0.32 (0.03–3.56)	0.3547
I don’t think so	23	36.51%	29	46.03%	11	17.46%		0.35 (0.04–3.16)	0.3484	0.12 (0.01–1.20)	0.0711
never	9	40.91%	9	40.91%	4	18.18%		0.29 (0.03–2.91)	0.2919	0.11 (0.01–1.34)	0.0834

N—number of respondents; %—percent; *p* #—statistical significance of the Chi square test; *p* *—statistical significance of univariate logistic regression; OR—odds ratio; 95% CI—95% confidence interval; N/A—Not Applicable.

**Table 4 ijerph-19-09802-t004:** Unfavorable emotional behavior and the risk of sleep disorders in nurses.

Athens Insomnia Scale (AIS)							*p* #	Insomnia Risk + Clinical Insomnia (vs. Norm)		Clinical Insomnia (vs. Norm)	
	Norm		Insomnia Risk		Clinical Insomnia						
	N	%	N	%	N	%					
Do you experience mood changes such as frustration after your night shift?								OR (95% CI)	*p* *	OR (95% CI)	*p* *
No	38	35.85%	46	43.40%	22	20.75%	0.0462	1.00	-	1.00	-
Yes	2	10.00%	12	60.00%	6	30.00%		5.03 (1.11–22.86)	0.0365	5.18 (0.96–27.92)	0.0556
depression?											
No	39	32.50%	57	47.50%	24	20.00%	0.0544	1.00	-	1.00	-
Yes	1	16.67%	1	16.67%	4	66.67%		2.41 (0.27–21.31)	0.4298	6.50 (0.69–61.64)	0.1029
hostility?											
No	39	32.23%	56	46.28%	26	21.49%	0.6359	1.00	-	1.00	-
Yes	1	20.00%	2	40.00%	2	40.00%		1.90 (0.21–17.59)	0.5709	3.00 (0.26–34.81)	0.3797
helplessness?											
No	39	34.21%	53	46.49%	22	19.30%	0.0344	1.00	-	1.00	-
Yes	1	8.33%	5	41.67%	6	50.00%		5.72 (0.71–45.94)	0.1009	10.64 (1.20–94.15)	0.0336
annoyance?											
No	29	40.28%	31	43.06%	12	16.67%	0.0357	1.00	-	1.00	-
Yes	11	20.37%	27	50.00%	16	29.63%		2.64 (1.17–5.94)	0.0194	3.52 (1.27–9.76)	0.0158
irritation?											
No	36	36.36%	45	45.45%	18	18.18%	0.0350	1.00	-	1.00	-
Yes	4	14.81%	13	48.15%	10	37.04%		3.29 (1.06–10.25)	0.0405	5.00 (1.38–18.67)	0.0145
nervousness?											
No	31	35.23%	39	44.32%	18	20.45%	0.4150	1.00	-	1.00	-
Yes	9	23.68%	19	50.00%	10	26.32%		1.75 (0.74–4.17)	0.2044	1.91 (0.66–5.59)	0.2352
feeling anxious?											
No	38	33.63%	54	47.79%	21	18.58%	0.0276	1.00	-	1.00	-
Yes	2	15.38%	4	30.77%	7	53.85%		2.79 (0.59–13.21)	0.1968	6.33 (1.21–33.29)	0.0292

N—number of respondents; %—percent; *p* #—statistical significance of the Chi square test; *p* *—statistical significance of univariate logistic regression; OR—odds ratio; 95% CI—95% confidence interval.

**Table 5 ijerph-19-09802-t005:** Disorders of the gastrointestinal tract and the occurrence of sleep disorders.

Athens Insomnia Scale (AIS)							*p* #	Insomnia Risk + Clinical Insomnia (vs. Norm)		Clinical Insomnia (vs. Norm)	
	Norm		Insomnia Risk		Clinical Insomnia						
	N	%	N	%	N	%					
Do you suffer from any digestive problems, such as constipation?								OR (95% CI)	*p* *	OR (95% CI)	*p* *
No	31	34.07%	42	46.15%	18	19.78%	0.4925	1.00	-	1.00	-
Yes	9	25.71%	16	45.71%	10	28.57%		1.49 (0.62–3.57)	0.3687	1.91 (0.66–5.59)	0.2352
nausea?											
No	38	33.04%	54	46.96%	23	20.00%	0.1839	1.00	-	1.00	-
Yes	2	18.18%	4	36.36%	5	45.45%		2.22 (0.46–10.79)	0.3225	4.13 (0.74–23.06)	0.1060
heartburn?											
No	34	37.36%	41	45.05%	16	17.58%	0.0357	1.00	-	1.00	-
Yes	6	17.14%	17	48.57%	12	34.29%		2.88 (1.09–7.65)	0.0335	4.25 (1.35–13.37)	0.0133
diarrhea?											
No	37	31.90%	56	48.28%	23	19.83%	0.0875	1.00	-	1.00	-
Yes	3	30.00%	2	20.00%	5	50.00%		1.09 (0.27–4.47)	0.9016	2.68 (0.59–12.30)	0.2044
flatulence?											
No	28	40.00%	30	42.86%	12	17.14%	0.0588	1.00	-	1.00	-
Yes	12	21.43%	28	50.00%	16	28.57%		2.44 (1.10–5.43)	0.0281	3.11 (1.14–8.53)	0.0274
epigastric pain?											
No	33	33.67%	43	43.88%	22	22.45%	0.6112	1.00	-	1.00	-
Yes	7	25.00%	15	53.57%	6	21.43%		1.52 (0.59–3.95)	0.866	1.29 (0.38–4.34)	0.6856
I do not suffer from any of the aforementioned ailments from the digestive system											
No	22	25.00%	44	50.00%	22	25.00%	0.0493	1.00	-	1.00	-
Yes	18	47.37%	14	36.84%	6	15.79%		0.37 (0.17–0.82)	0.0148	0.33 (0.11–0.99)	0.0496

N—number of respondents; %—percent; *p* #—statistical significance of the Chi square test; *p* *—statistical significance of univariate logistic regression; OR—odds ratio; 95% CI—95% confidence interval.

**Table 6 ijerph-19-09802-t006:** Disorders of the cardiovascular system and the risk of sleep disorders.

Athens Insomnia Scale (AIS)							*p* #	Insomnia Risk + Clinical Insomnia (vs. Norm)		Clinical Insomnia (vs. Norm)	
	Norm		Insomnia Risk		Clinical Insomnia						
	N	%	N	%	N	%					
Do you have any problems with the cardiovascular system, such as palpitations?								OR (95% CI)	*p* *	OR (95% CI)	*p* *
No	30	37.97%	35	44.30%	14	17.72%	0.0931	1.00	-	1.00	-
Yes	10	21.28%	23	48.94%	14	29.79%		2.65 (0.99–5.21)	0.0544	3.00 (1.07–8.40)	0.0366
irregular heartbeat?											
No	37	33.64%	50	45.45%	23	20.91%	0.4087	1.00	-	1.00	-
Yes	3	18.75%	8	50.00%	5	31.25%		2.20 (0.59–8.19)	0.2414	2.68 (0.59–12.30)	0.2044
elevated BP values?											
No	30	30.93%	48	49.48%	19	19.59%	0.2908	1.00	-	1.00	-
Yes	10	34.48%	10	34.48%	9	31.03%		0.85 (0.35–2.05)	0.7184	1.42 (0.49–4.14)	0.5191
increased heart rate?											
No	27	33.75%	40	50.00%	13	16.25%	0.1099	1.00	-	1.00	-
Yes	13	28.26%	18	39.13%	15	32.61%		1.29 (0.59–2.85)	0.5244	0.40 (0.89–6.48)	0.0850
I do not suffer from any of the aforementioned cardiovascular complaints											
No	25	28.41%	40	45.45%	23	26.14%	0.1993	1.00	-	1.00	-
Yes	15	39.47%	18	47.37%	5	13.16%		0.61 (0.27–1.35)	0.2227	0.36 (0.11–1.16)	0.0862

N—number of respondents; %—percent; *p* #—statistical significance of the Chi square test; *p* *—statistical significance of univariate logistic regression; OR—odds ratio; 95% CI—95% confidence interval.

**Table 7 ijerph-19-09802-t007:** The dependence of changes in behavior and the manner of responding respondents and the risk of sleep disorders.

Athens Insomnia Scale (AIS)							*p* #	Insomnia Risk + Clinical Insomnia (vs. Norm)		Clinical Insomnia (vs. Norm)	
	Norm		Insomnia Risk		Clinical Insomnia						
	N	%	N	%	N	%					
Do you experience any changes in mood after your night shift?								OR (95% CI)	*p* *	OR (95% CI)	*p* *
No	21	50.00%	16	38.10%	5	11.90%	0.0053	1.00	-	1.00	-
Yes	19	22.62%	42	50.00%	23	27.38%		3.42 (1.55—7.55)	0.0023	5.08 (1.61–16.04)	0.0055
Have you noticed that you are constantly tired?											
No	38	41.30%	45	48.91%	9	9.78%	<0.0001	1.00	-	1.00	-
Yes	2	5.88%	13	38.24%	19	55.88%		11.26 (2.51–49.84)	0.0014	40.11 (7.87–204.35)	<0.0001
Have you noticed the presence of aggression by yourself?											
No	38	33.63%	54	47.79%	21	18.58%	0.0274	1.00	-	1.00	-
Yes	2	15.38%	4	30.77%	7	53.85%		2.79 (0.59–13.21)	0.1968	6.33 (1.21–33.29)	0.0292
Have you noticed any criticism?											
No	37	35.24%	50	47.62%	18	17.14%	0.0101	1.00	-	1.00	-
Yes	3	14.29%	8	38.10%	10	47.62%		3.27 (0.90–11.82)	0.0714	6.85 (1.68–28.00)	0.0074
Have you noticed any exaggeration in the way you react?											
No	36	34.29%	52	49.52%	17	16.19%	0.0033	1.00	-	1.00	-
Yes	4	19.05%	6	28.57%	11	52.38%		2.22 (0.69–7.08)	0.1790	5.82 (1.62–20.98)	0.0070
Do you have anxiety after the night shift?											
No	37	32.74%	56	49.56%	20	17.70%	0.0012	1.00	-	1.00	-
Yes	3	23.08%	2	15.38%	8	61.54%		1.62 (0.42–6.25)	0.4817	4.93 (1.18–20.70)	0.0292

N—number of respondents; %—percent; *p* #—statistical significance of the Chi square test; *p* *—statistical significance of univariate logistic regression; OR—odds ratio; 95% CI—95% confidence interval.

**Table 8 ijerph-19-09802-t008:** The relationship between the way of spending free time and the risk of sleep disorders in the surveyed nurses.

Athens Insomnia Scale (AIS)							*p* #	Insomnia Risk + Clinical Insomnia (vs. Norm)		Clinical Insomnia (vs. Norm)	
	Norm		Insomnia Risk		Clinical Insomnia						
	N	%	N	%	N	%					
I spend my free time in gymnastics								OR (95% CI)	*p* *	OR (95% CI)	*p* *
No	28	26.17%	54	50.47%	25	23.36%	0.0074	1.00	-	1.00	-
Yes	12	63.16%	4	21.05%	3	15.79%		0.21 (0.07–0.58)	0.0026	0.28 (0.07–1.11)	0.0697
I spend my free time reading											
No	16	23.19%	40	57.97%	13	18.84%	0.0102	1.00	-	1.00	-
Yes	24	42.11%	18	31.58%	15	26.32%		0.42 (0.19–0.89)	0.0247	0.77 (0.29–2.04)	0.5981

N—number of respondents; %—percent; *p* #—statistical significance of the Chi square test; *p* *—statistical significance of univariate logistic regression; OR—odds ratio; 95% CI—95% confidence interval.

**Table 9 ijerph-19-09802-t009:** The impact of shift work on the family life of respondents and the problems of sleep disorders.

Athens Insomnia Scale (AIS)							*p* #	Insomnia Risk + Clinical Insomnia (vs. Norm)		Clinical Insomnia (vs. Norm)	
	Norm		Insomnia Risk		Clinical Insomnia						
	N	%	N	%	N	%					
Do you think that shift work has an impact on family life?								OR (95% CI)	*p* *	OR (95% CI)	*p* *
Moderately negative	20	30.77%	33	50.77%	12	18.46%	0.1770	1.00	-	1.00	-
Moderately positive	1	33.33%	2	66.67%	0	0.00%		0.89 (0.08–10.38)	0.9252	N/A	N/A
Definitely negative	10	23.81%	19	45.24%	13	30.95%		1.42 (0.59–3.44)	0.4349	2.17 (0.73–6.46)	0.1651
Has no effect	9	56.25%	4	25.00%	3	18.75%		0.35 (0.11–1.06)	0.0629	0.56 (0.13–2.47)	0.4394
Do you think that working in a shift system has an impact on lowering sexual activity?											
Yes	20	28.17%	33	46.48%	18	25.35%	0.5184	1.00	-	1.00	-
No	15	42.86%	14	40.00%	6	17.14%		0.52 (0.22–1.22)	0.1330	0.44 (0.14–1.39)	0.1636
I have no opinion	5	25.00%	11	55.00%	4	20.00%		1.18 (0.38–3.67)	0.7793	0.89 (0.21–3.83)	0.8744
Do you think that working in a shift system has an impact on lowering immunity and the occurrence of more frequent infections?											
Yes, several times a year	18	25.35%	35	49.30%	18	25.35%	0.3660	1.00	-	1.00	-
Yes, once a year	7	29.17%	13	54.17%	4	16.67%		0.83 (0.29–2.31)	0.7139	0.57 (0.14–2.30)	0.4305
No	6	46.15%	5	38.46%	2	15.38%		0.40 (0.12–1.34)	0.1352	0.33 (0.06–1.88)	0.2129
I do not know	9	50.00%	5	27.78%	4	22.22%		0.34 (0.12–0.99)	0.0474	0.44 (0.12–1.71)	0.2380
Do you think that shift work has an impact on social life?											
Moderately negative	21	37.50%	25	44.64%	10	17.86%	0.2449	1.00	-	1.00	-
Moderately positive	1	50.00%	1	50.00%	0	0.00%		0.72 (0.04–10.11)	0.7230	N/A	N/A
Definitely negative	12	25.00%	20	41.67%	16	33.33%		1.80 (0.77–4.20)	0.1744	2.80 (0.97–8.10)	0.0573
Has no effect	6	30.00%	12	60.00%	2	10.00%		1.40 (0.47–4.20)	0.5484	0.70 (0.12–4.10)	0.6927
Do you drink coffee/strong tea?											
No	3	23.08%	4	30.77%	6	46.15%	0.1258	1.00	-	1.00	-
Yes	37	32.74%	54	47.79%	22	19.47%		0.62 (0.16–2.37)	0.4817	0.30 (0.07–1.31)	0.1089
Do you see a relationship with the amount of coffee/strong tea drunk with night duty?											
No	28	40.00%	29	41.43%	13	18.57%	0.0755	1.00	-	1.00	-
Yes	12	21.43%	29	51.79%	15	26.79%		2.44 (1.10–5.43)	0.0281	2.69 (0.99–7.35)	0.0533
Do you smoke cigarettes?											
Yes	6	37.50%	6	37.50%	4	0.25	0.2324	1.00	-	1.00	-
I was smoking but currently not	1	8.33%	6	50.00%	5	0.4167		6.6 (0.67–64.77)	0.1053	7.50 (0.62–90.65)	0.1130
I do not smoke	33	33.67%	46	46.94%	19	0.1939		0.18 (0.40–3.53)	0.7650	0.86 (0.22–3.45)	0.8357
Do you see the relationship between the number of cigarettes smoked on days and the night duty?											
Yes	2	12.50%	9	56.25%	5	0.3125	0.3249	1.00	-	1.00	-
No	8	36.36%	8	36.36%	6	0.2727		0.25 (0.05–1.39)	0.1136	0.30 (0.04–2.11)	0.2267
Not applicable	30	34.09%	41	46.59%	17	0.1932		0.28 (0.06–1.30)	0.1028	0.23 (0.04–1.30)	0.0954
Do you drink alcohol?											
Several times a month	9	39.13%	9	39.13%	5	21.74%	0.4685	1.00	-	1.00	-
Occasionally	22	29.33%	39	52.00%	14	18.67%		1.55 (0.59–4.10)	0.3787	1.15 (0.32–4.13)	0.8356
I do not drink	9	32.14%	10	35.71%	9	32.14%		1.36 (0.43–4.30)	0.6038	1.80 (0.43–7.53)	0.4209
Is there a relationship between the amount of alcohol drunk and shift work?											
Yes	1	10.00%	6	60.00%	3	30.00%	0.4741	1.00	-	1.00	-
No	23	31.08%	35	47.30%	16	21.62%		0.25 (0.03–2.06)	0.1961	0.23 (0.02–2.44)	0.2231
Not applicable	16	38.10%	17	40.48%	9	21.43%		0.18 (0.02–1.56)	0.1200	0.19 (0.02–2.08)	0.1727
Do you drink high-energy fluids?											
Several times a month	3	25.00%	6	50.00%	3	25.00%	0.8640	1.00	-	1.00	-
I do not drink	37	32.46%	52	45.61%	25	21.93%		0.69 (0.18–2.71)	0.5993	0.68 (0.13–6.62)	0.6472
Do you see a relationship with the amount of high-energy fluids drunk with night duty?											
Yes	1	14.29%	4	57.14%	2	28.57%	0.7462	1.00	-	1.00	-
No	10	30.30%	17	51.52%	6	18.18%		0.38 (0.04–3.61)	0.4022	0.30 (0.02–4.06)	0.3650
Not applicable	29	33.72%	37	43.02%	20	23.26%		0.33 (0.04–2.85)	0.3121	0.35 (0.03–407)	0.3976
Do you take any drugs to help you fall asleep?											
No	40	36,70%	51	46,79%	18	16,51%	<0.0001	1.00	-	1.00	-
Yes	0	0.00%	7	41,18%	10	58,82%		N/A	N/A	N/A	N/A

N—number of respondents; %—percent; *p* #—statistical significance of the Chi square test; *p* *—statistical significance of univariate logistic regression; OR—odds ratio; 95% CI—95% confidence interval; N/A—Not Applicable.

**Table 10 ijerph-19-09802-t010:** Caring for one’s own appearance and respondents’ sleep disorders.

Athens Insomnia Scale (AIS)							*p* #	Insomnia Risk + Clinical Insomnia (vs. Norm)		Clinical Insomnia (vs. Norm)	
	Norm		Insomnia Risk		Clinical Insomnia						
	N	%	N	%	N	%					
In your own assessment, are you a conflict person?								OR (95% CI)	*p* *	OR (95% CI)	*p* *
Yes	1	20.00%	2	40.00%	2	40.00%	0.0774	1.00	-	1.00	-
No	36	36.73%	45	45.92%	17	17.35%		0.48 (0.20–1.16)	0.1025	0.24 (0.02–2.79)	0.2518
I do not know	3	13.04%	11	47.83%	9	39.13%		1.86 (0.62–5.84)	0.2675	1.50 (0.10–23.07)	0.7712
Do you take care of your appearance?											
Yes, always	32	37.65%	40	47.06%	13	15.29%	0.0331	1.00	-	1.00	-
Sometimes I forget about my appearance	8	20.00%	18	45.00%	14	35.00%		0.02 (0.01–0.03)	< 0.0001	4.31 (1.46–12.71)	0.0081
No	0	0.00%	0	0.00%	1	100.00%		N/A	N/A	N/A	N/A
Have you noticed any weight gain?											
Yes	24	32.88%	27	36.99%	22	30.14%	0.0149	1.00	-	1.00	-
No	16	30.19%	31	58.49%	6	11.32%		1.06 (0.73–1.56)	0.7490	0.41 (0.14–0.23)	0.1120

N—number of respondents; %—percent; *p* #—statistical significance of the Chi square test; *p* *—statistical significance of univariate logistic regression; OR—odds ratio; 95% CI—95% confidence interval; N/A—Not Applicable.

**Table 11 ijerph-19-09802-t011:** Menstrual disorders and sleep disorders in nurses (analysis made only for women).

Athens Insomnia Scale (AIS)							*p* #	Insomnia Risk + Clinical Insomnia (vs. Norm)		Clinical Insomnia (vs. Norm)	
	Norm		Insomnia Risk		Clinical Insomnia						
	N	%	N	%	N	%					
Do you have any menstrual disorders?								OR (95% CI)	*p* *	OR (95% CI)	*p* *
No	20	27.78%	39	54.17%	13	18.06%	0.1047	1.00	-	1.00	-
Yes	11	26.83%	17	41.46%	13	31.71%		1.05 (0.44–2.84)	0.9135	1.82 (0.63–5.27)	0.2708
Not applicable	6	60.00%	2	20.00%	2	20.00%		0.26 (0.06–1.01)	0.0509	0.51 (0.09–2.94)	0.4535
Menstruation is painful											
No	31	35.23%	41	46.59%	16	18.18%	0.0578	1.00	-	1.00	-
Yes	6	17.14%	17	48.57%	12	34.29%		2.63 (0.99–7.02)	0.0537	3.88 (1.23–12.45)	0.0211
Menstruation is profuse											
No	34	37.78%	41	45.56%	15	16.67%	0.0012	1.00	-	1.00	-
Yes	3	9.09%	17	51.52%	13	39.39%		6.07 (1.72–21.43)	0.0051	9.82 (2.44–39.62)	0.0013
Menstruation is tight											
No	33	30.28%	49	44.95%	27	24.77%	0.2069	1.00	-	1.00	-
Yes	4	28.57%	9	64.29%	1	7.14%		1.09 (0.32–3.71)	0.8959	0.31 (0.03–2.90)	0.3016
The menstruation is fine											
No	19	24.05%	37	46.84%	23	29.11%	0.0311	1.00	-	1.00	-
Yes	18	40.91%	21	47.73%	5	11.36%		0.46 (0.21–1.01)	0.0529	0.23 (0.07–0.73)	0.0131
Menstruation is irregular											
No	29	30.85%	48	51.06%	17	18.09%	0.0873	1.00	-	1.00	-
Yes	8	27.59%	10	34.48%	11	37.93%		1.17 (0.47–3.00)	0.7377	2.35 (0.79–6.98)	0.1252
Did you have a spontaneous miscarriage during the years of shift work?											
No	34	32.08%	51	48.11%	21	19.81%	0.1509	1.00	-	1.00	-
Yes	3	17.65%	7	41.18%	7	41.18%		2.20 (0.59–8.18)	0.2378	3.78 (0.88–16.23)	0.0739

N—number of respondents; %—percent; *p* #—statistical significance of the Chi square test; *p* *—statistical significance of univariate logistic regression; OR—odds ratio; 95% CI—95% confidence interval.

**Table 12 ijerph-19-09802-t012:** Body functioning disorders and sleep disorders in nurses.

Athens Insomnia Scale (AIS)							*p* #	Insomnia Risk + Clinical Insomnia (vs. Norm)		Clinical Insomnia (vs. Norm)	
	Norm		Insomnia Risk		Clinical Insomnia						
	N	%	N	%	N	%					
Does your shift work cause a negative attitude?								OR (95% CI)	*p* *	OR (95% CI)	*p* *
No	29	39.19%	33	44.59%	12	16.22%	0.0446	1.00	-	1.00	-
Yes	11	21.15%	25	48.08%	16	30.77%		1.55 (1.03–2.33)	0.0346	3.52 (1.27–9.76)	0.0158
Does your shift work result in a loss of objectivity?											
No	33	32.04%	47	45.63%	23	22.33%	0.9813	1.00	-	1.00	-
Yes	7	30.43%	11	47.83%	5	21.74%		1.04 (0.64–1.69)	0.8812	1.03 (0.29–3.63)	0.9697
Does your shift work result in an inability to concentrate and make decisions?											
No	19	26.03%	41	56.16%	13	17.81%	0.0261	1.00	-	1.00	-
Yes	21	39.62%	17	32.08%	15	28.30%		0.73 (0.50–1.07)	0.1075	1.04 (0.40–2.75)	0.9306
According to you, does shift work cause problems with memory?											
No	35	39.33%	41	46.07%	13	14.61%	0.0011	1.00	-	1.00	-
Yes	5	13.51%	17	45.95%	15	40.54%		2.04 (1.21–3.42)	0.0070	8.08 (2.44–26.70)	0.0006
Does your shift work increase your susceptibility to accidents?											
No	31	38.75%	33	41.25%	16	20.00%	0.0747	1.00	-	1.00	-
Yes	9	19.57%	25	54.35%	12	26.09%		1.61 (1.052.47)	0.0287	2.83 (0.90–7.41)	0.0776
Does your shift work cause more frequent mistakes?											
No	32	39.02%	35	42.68%	15	18.29%	0.0403	1.00	-	1.00	-
Yes	8	18.18%	23	52.27%	13	29.55%		1.70 (1.09–2.64)	0.0192	3.47 (1.19–10.14)	0.0232
Are you under the care of an ophthalmologist?											
No	26	40.63%	30	46.88%	8	12.50%	0.0111	1.00	-	1.00	-
Yes	14	22.58%	28	45.16%	20	32.26%		1.53 (1.04–2.26)	0.0314	4.64 (1.63–13.22)	0.0040
Do you wear glasses?											
No	24	35.82%	31	46.27%	12	17.91%	0.3768	1.00	-	1.00	-
Yes	16	27.12%	27	45.76%	16	27.12%		1.26 (0.84–1.79)	0.2962	2.00 (0.75–5.33)	0.1657
Do you use eye moisturizers?											
No	35	35.35%	42	42.42%	22	22.22%	0.1853	1.00	-	1.00	-
Yes	5	18.52%	16	59.26%	6	22.22%		1.55 (0.92–2.63)	0.1028	1.91 (0.52–7.01)	0.3300
Do you have headaches, migraines?											
No	22	44.90%	17	34.69%	10	20.41%	0.0354	1.00	-	1.00	-
Yes	18	23.38%	41	53.25%	18	23.38%		1.63 (1.11–2.40)	0.0126	2.20 (0.82–5.94)	0.1196
Are you treated neurologically?											
No	35	30.43%	53	46.09%	27	23.48%	0.4027	1.00	-	1.00	-
Yes	5	45.45%	5	45.45%	1	9.09%		0.73 (0.39–1.36)	0.3129	0.26 (0.03–2.35)	0.2302
Are you under neurological treatment because of your spine?											
No	36	30.51%	55	46.61%	27	22.88%	0.5111	1.00	-	1.00	-
Yes	4	50.00%	3	37.50%	1	12.50%		0.66 (0.32–1.36)	0.2626	0.33 (0.04–3.15)	0.3380
Are you being treated neurologically because of migraine?											
No	39	31.97%	56	4.59%	27	22.13%	0.9554	1.00	-	1.00	-
Yes	1	25.00%	2	50.00%	1	25.00%		1.19 (038–3.74)	0.7694	1.44 (0.09–24.11)	0.7979
Are you undergoing neurological treatment because of your shoulder?											
No	40	32.26%	57	45.97%	27	21.77%	0.4048	1.00	-	1.00	-
Yes	0	0.00%	1	50.00%	1	50.00%		N/A	N/A	N/A	N/A

N—number of respondents; %—percent; *p* #—statistical significance of the Chi square test; *p* *—statistical significance of univariate logistic regression; OR—odds ratio; 95% CI—95% confidence interval; N/A—Not Applicable.

**Table 13 ijerph-19-09802-t013:** Results of multivariate logistic regression performed in a group of female participants.

Variable		Insomnia Risk + Clinical Insomnia (vs. Norm)	
		OR (95% CI)	*p* ^
Have you noticed that you are constantly tired?	No	1.00	-
	Yes	10.22 (1.68–62.02)	0.0115
Menstruation is profuse	No	1.00	-
	Yes	5.40 (1.03–28.28)	0.0460
Do you take care of your appearance?	Yes, always	1.00	-
	Sometimes I forget about my appearance	8.23 (1.98–34.25)	0.0038
	No	N/A	N/A
Do you ever feel sleepy at work?	Several times a month	1.00	-
	A few times a week	0.30 (0.05–1.88)	0.1973
	Occasionally	0.12 (0.03–0.45)	0.0015
	Never	0.04 (0.01–1.40)	0.0758
I spend my free time reading	No	1.00	-
	Yes	0.13 (0.04–0.43)	0.0008
Variable		Clinical Insomnia (vs. Norm)	
		OR (95% CI)	*p* ^
Have you noticed that you are constantly tired?	No	1.00	-
	Yes	9.81 (1.52–63.34)	0.0164
Menstruation is profuse	No	1.00	-
	Yes	20.37 (1.59–261.67)	0.0207
Are you under the care of an ophthalmologist?	No	1.00	-
	Yes	19.72 (1.90–204.862)	0.0125
According to you, does shift work cause problems with memory?	No	1.00	-
	Yes	21.26 (1.73–259.542)	0.0170

*p* ^—statistical significance of multivariate logistic regression; OR—odds ratio; 95% CI—95% confidence interval; N/A—Not Applicable.

**Table 14 ijerph-19-09802-t014:** Results of multivariate logistic regression carried out among all participants.

Variable		Insomnia Risk + Clinical Insomnia (vs. Norm)	
		OR (95% CI)	*p* ^
Have you noticed that you are constantly tired?	No	1.00	-
	Yes	3.40 (1.03–11.17)	0.0442
Are you under the care of an ophthalmologist?	No	1.00	-
	Yes	3.39 (1.16–9.88)	0.0255
According to you. does shift work cause problems with memory?	No	1.00	-
	Yes	4.40 (1.08–18.00)	0.0391
Does your shift work increase your susceptibility to accidents?	No	1.00	-
	Yes	4.73 (1.43–9.88)	0.0255
Do you ever feel sleepy at work?	Several times a month	1.00	-
	A few times a week	0.26 (0.04–1.54)	0.1375
	Occasionally	0.20 (0.06–0.65)	0.0071
	Never	NA	NA
I spend my free time reading	No	1.00	-
	Yes	0.19 (0.06–0.58)	0.0214
I spend my free time in gymnastics	No	1.00	-
	Yes	0.20 (0.06–0.69)	0.0109
Variable		Clinical Insomnia (vs. Norm)	
		OR (95% CI)	*p* ^
Are you under the care of an ophthalmologist?	No	1.00	-
	Yes	54.29 (4.98–591.68)	0.0010
According to you. does shift work cause problems with memory?	No	1.00	-
	Yes	38.88 (3.56–424.57)	0.0027
Do you have any problems with the cardiovascular system, such as palpitations?	No	1.00	-
	Yes	14.25 (1.87–108.39)	0.0103
Do you ever feel sleepy at work?	Several times a month	1.00	-
	A few times a week	0.90 (0.10–8.20)	0.9235
	Occasionally	0.06 (0.01–0.49)	0.0080
	Never	N/A	N/A

*p* ^—statistical significance of multivariate logistic regression; OR—odds ratio; 95% CI—95% confidence interval; N/A—Not Applicable.

## Data Availability

All data used in the publication are available with Małgorzata Knap.

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
