# Peer review of "Sleep Disturbances and Health Consequences Induced by the Specificity of Nurses’ Work"

_ijerph, 2022, doi:10.3390/ijerph19169802_

Round 1
Reviewer 1 Report
- The Sample size calculations are not given. Please explain why 126 is sufficient to draw rubust conclusions
- the Authors perfomed a descriptive analysis only. They used the chi-square test, but probably in many cases the Fisher exact test or the Yates correction must be considered due to small numbers in the cells
- There are alot of tables. The Authors can marge some of these tables in order to better present the results
- A multivariate analysis is missing. I suggest to use a logistic regression analysis using as a dependent variable the "clinical insomnia" in the first model, and the sum of "clinical insomnia" and "Insomnia risk" in a second model.
- In tables 2 and 12 some percentages are not reported correctly
- some indications to improve sleep quality among healthcare professional must be given. See
doi: 10.7417/CT.2022.2414
doi: 10.1016/j.ijnurstu.2019.103513
doi: 10.1016/j.apnr.2019.151227
doi: 10.1708/3790.37738
- The limts of the study must be acknowledged in the Discussion section
Author Response
Dear Reviewer,
We want to express our great appreciation to You and the reviewers for taking the time and effort necessary to review our manuscript entitled: Sleep disturbances and health consequences induced by the specificity of nurses' work
We carefully considered your comments. They helped us a lot; we appreciate Your patience and willingness to help us to make this manuscript better. Herein, we explain how we revised the paper for a second time based on Your comments and recommendations. All changes are listed in the file below. We have accepted all of your suggestions. In table toy can see our answers and also there are a red text.
Yours sincerely
Sabina Krupa
|
COMMENTS FROM THE EDITOR |
CHANGES MADE |
|
The Sample size calculations are not given. Please explain why 126 is sufficient to draw rubust conclusions |
Thank you very much for this suggestion. The sample size was calculated prior to conducting the surveys. It was given at work. |
|
the Authors perfomed a descriptive analysis only. They used the chi-square test, but probably in many cases the Fisher exact test or the Yates correction must be considered due to small numbers in the cells |
Thank you very much for this suggestion. Yates' correction or Fisher's test is applied to 2x2 tables, when the condition with sufficiently large expected numbers is not met (most often it is assumed to be 5, possibly 10 for the observed numbers). Cochran also gave the rule that at most 20% of the expected numbers may be less than 5. The Yates correction, called the discontinuity correction, is applied to the 2x2 tables, because only then there is a problem with the discrepancy between the true discrete distribution and the continuous chi ^ 2 distribution, which we use to approximate the distribution of the test statistic and from which we calculate the p-value. With larger tables, there are more values and the real distributions of statistics are close to the continuous distribution. It is not recommended to use the Yates correction then, because the regular chi ^ 2 test gives more close to the real values.
Yates F. Contingency Tables Involving Small Numbers and the χ2 Test. Supplement to the Journal of the Royal Statistical Society , 1934, Vol. 1, No. 2 (1934), pp. 217-235 |
|
There are alot of tables. The Authors can marge some of these tables in order to better present the results |
|
|
A multivariate analysis is missing. I suggest to use a logistic regression analysis using as a dependent variable the "clinical insomnia" in the first model, and the sum of "clinical insomnia" and "Insomnia risk" in a second model. |
Thank you very much for this comment. The analysis was carried out and its results were added to the tables. |
|
In tables 2 and 12 some percentages are not reported correctly |
We chenged it |
|
some indications to improve sleep quality among healthcare professional must be given. See doi: 10.7417/CT.2022.2414, doi: 10.1016/j.ijnurstu.2019.10351, doi: 10.1016/j.apnr.2019.151227, doi: 10.1708/3790.37738 |
We added it |
|
The limts of the study must be acknowledged in the Discussion section
|
We added informations: Shift work has negative health consequences in the group of nurses. Most of the subjects suffered from sleep problems, neurological problems, and social problems. All these elements were related to night work. The respondents also complained about problems related to the gastrointestinal tract. Another problem of night nurses is the risk of de-veloping cardiovascular diseases. |

Reviewer 2 Report
The Authors decided to analyze the problem of insomnia among the night shift workers. This problem has already been a subject of the numerous studies, however additional data from Polish environment would be still very valuable. None the less I would like to express my doubts about the submitted manuscript.
Overall, the quality of the manuscript is low. It needs English language and punctuation revision.
The authors use two names “Athenian” and “Athens” insomnia scale. Please use the proper one and be consistent throughout the whole manuscript. What is more, the Authors after introducing the abbreviation of this scale still use the full name in numerous places. Please correct it.
Line 171 – this data is presented in the table 3, not 2.
Line 186 – this data is presented in the table 4, not 3.
Line 208 – this data is presented in the table 5, not 4.
Line 223 – this data is presented in the table 6, not 5.
Line 237 – this data is presented in the table 7, not 6.
My main doubt refers to the conclusions presented by the Authors.
The authors analyzed the data comparing the occurrence of different conditions (emotional and physical symptoms) between the groups at different insomnia risk. At the same time, the number of the nights shift did not correlate with the insomnia risk. That is why I suggest analyzing the occurrence of the mentioned above conditions referring to the load of night shifts and other aspects of nurses’ work (work experience etc.). The presented analysis shows the correlation between the insomnia risk with the selected symptoms, but it does not directly prove that it is related to nurses’ work.
Line 402 – The study did not prove that. The study shown that there is no correlation between the number of the night shifts and the insomnia risk. The correlation was found only for the work experience, which is not equal to the night shifts load (nurses could work on day shift or in other facilities throughout these years). Secondary, the correlations were then made between the insomnia risk (not the work aspects) and the selected symptoms and life aspects, that is why these conclusions do not match the results. Please provide additional analysis to support these conclusions.
Other major comments:
Line 84 – please elaborate on the study design
Please provide information how were the questionnaires administered.
Please provide the full questionnaire use (e.g. in Supplementary materials).
Line 86 – please provide calculation of the needed sample size
Table 1 – What does p value refers to? Is this the comparison only between the age groups (number of shifts etc.) within the same insomnia risk subgroup or between the insomnia risk subgroups, or both? P value should be gives for each subgroup analysis.
Table 2 – Similar as above: what does p value refers to – insomnia risk subgroup comparison or other subcategories comparison? Please make it clear.
Table 11. It seems that the menstruation cycle questions included into the analysis gathered also from male participants, which could interfere the results. Please explain.
The Authors presented the Discussion section as the pointing out the results of the works from other authors and their own, without any further commentary regarding the similarities and differences between the results. Please elaborate.
Lines 136-140 – Did the Authors asked the participants whether they take any medications that could interfere with the sleep quality or cause dizziness (apart from sleeping pills)?
There are also few minor comments.
Line 61 – please give the examples of such diseases
Line 80 – please specify the investigated consequences
Lines 119-135 – I suggest presenting this information in the table
Line 341 – please provide the reference.
To sum up, regarding my numerous doubts about the manuscript I suggest the revision of the study design and the manuscript and resubmission after essential improvements.
Author Response
Dear Reviewer,
We want to express our great appreciation to You and the reviewers for taking the time and effort necessary to review our manuscript entitled: Sleep disturbances and health consequences induced by the specificity of nurses' work
We carefully considered your comments. They helped us a lot; we appreciate Your patience and willingness to help us to make this manuscript better. Herein, we explain how we revised the paper for a second time based on Your comments and recommendations. All changes are listed in the file below. We have accepted all of your suggestions. In table toy can see our answers and also there are a red text.
Yours sincerely
Sabina Krupa
|
COMMENTS FROM THE EDITOR |
CHANGES MADE |
|
The authors use two names “Athenian” and “Athens” insomnia scale. Please use the proper one and be consistent throughout the whole manuscript. |
Done |
|
the Authors after introducing the abbreviation of this scale still use the full name in numerous places. Please correct it. |
Done |
|
Line 171 – this data is presented in the table 3, not 2. Line 186 – this data is presented in the table 4, not 3. Line 208 – this data is presented in the table 5, not 4. Line 223 – this data is presented in the table 6, not 5. Line 237 – this data is presented in the table 7, not 6. |
We corrected it |
|
My main doubt refers to the conclusions presented by the Authors. The authors analyzed the data comparing the occurrence of different conditions (emotional and physical symptoms) between the groups at different insomnia risk. At the same time, the number of the nights shift did not correlate with the insomnia risk. That is why I suggest analyzing the occurrence of the mentioned above conditions referring to the load of night shifts and other aspects of nurses’ work (work experience etc.). The presented analysis shows the correlation between the insomnia risk with the selected symptoms, but it does not directly prove that it is related to nurses’ work. |
Thank you very much for this comment. Carrying out analyzes with an additional division, e.g. the number of night lobbies, would lead to a situation when very few groups of people would be analyzed, which would significantly reduce the power of statistical analyzes. |
|
Line 402 – The study did not prove that. The study shown that there is no correlation between the number of the night shifts and the insomnia risk. The correlation was found only for the work experience, which is not equal to the night shifts load (nurses could work on day shift or in other facilities throughout these years). Secondary, the correlations were then made between the insomnia risk (not the work aspects) and the selected symptoms and life aspects, that is why these conclusions do not match the results. Please provide additional analysis to support these conclusions. |
Thank you very much for this comment. This conclusion has been changed. |
|
Line 84 – please elaborate on the study design Please provide information how were the questionnaires administered. Please provide the full questionnaire use (e.g. in Supplementary materials). |
An original questionnaire and AIS were used to obtain the research material. Participation in the study was voluntary, data for the study were obtained anonymously. The respondents provided the required answers after prior consent to participate in the study, and they also received instructions on how to use the research tools. The next stage of the research was subjecting the collected research material to statistical analyzes in order to formulate conclusions from the research |
|
Line 86 – please provide calculation of the needed sample size |
Thank you very much for this suggestion. The sample size was calculated prior to conducting the surveys. It was given at work. |
|
Table 1 – What does p value refers to? Is this the comparison only between the age groups (number of shifts etc.) within the same insomnia risk subgroup or between the insomnia risk subgroups, or both? P value should be gives for each subgroup analysis. |
Thank you very much for this comment. In this example, chi square analysis is performed simultaneously for all subgroups. In the current version of the work, the results of univariate and multivariate logistic regression were added, which significantly supplements the results obtained in the study. |
|
Table 2 – Similar as above: what does p value refers to – insomnia risk subgroup comparison or other subcategories comparison? Please make it clear. |
Thank you very much for this comment. In this example, chi square analysis is performed simultaneously for all subgroups. In the current version of the work, the results of univariate and multivariate logistic regression were added, which significantly supplements the results obtained in the study. |
|
Table 11. It seems that the menstruation cycle questions included into the analysis gathered also from male participants, which could interfere the results. Please explain. |
Thank you very much for this comment. The results of the analyzes in table 11 have been corrected and are currently only the results for women. |
|
The Authors presented the Discussion section as the pointing out the results of the works from other authors and their own, without any further commentary regarding the similarities and differences between the results. Please elaborate. |
As a result of the study, it can be concluded that a significant number of heavy duty shifts (6 and more) may lead to the adaptation of changes in the circadian rhythm, which results in the lack of correlation between sleep disorders and a large number of night shifts |
|
Lines 136-140 – Did the Authors asked the participants whether they take any medications that could interfere with the sleep quality or cause dizziness (apart from sleeping pills)? |
The survey questionnaire included the use of stimulants (strong coffee, strong tea, high-energy drinks and smoking nicotine) and the frequency of using drugs to help you fall asleep. The questionnaire did not include questions about the use of other medications by the respondents. |
|
Line 61 – please give the examples of such diseases
|
SzymÅ„ska, Czechór and KÄ™dra revealed circulatory system problems in nurses in the night system. These were arrhythmias and increased blood pressure. The obtained results indicate the need to extend research in this area.
|
|
Line 80 – please specify the investigated consequences
|
The consequences are presented in the statistical analyzes.
|
|
Lines 119-135 – I suggest presenting this information in the table
|
Table with sociodemographic features attached in Attachment 1. |

Round 2
Reviewer 1 Report
some statements are not in English
"Możliwość poprawy snu niesie również 544
aktywność fizyczna. W swojej pracy Okechukwu et al., podkreślają, że przestrzeganie 545
programu ćwiczeÅ„ aerobowych o umiarkowanej intensywnoÅ›ci może poprawić zarówno 546
jakość, jak i czas trwania snu [23]. Z kolei badania Querstret et al. wykazały, że dłuższe 547
zmiany, w tym noce, oraz niewystarczający czas regeneracji między zmianami wiązały 548
się z gorszym snem, zwiększoną sennością i zwiększonym poziomem zmęczenia [24]. 549
WedÅ‚ug wielu badaÅ„, należy opracować interwencje dotyczÄ…ce snu, które mogÄ… 550
pozytywnie promować zdrowie pielęgniarek i sprzyjać efektywnej wydajności pracy 551
[25]. Ponadto decydenci szpitali powinni rozważyć opracowanie nowych wytycznych, 552
aby zminimalizować negatywny wpływ rotacji nocnej na zdrowie psychiczne i jakość 553
życia pielęgniarek [26]. 554
"
Author Response
Dear August 2022
Editor and Reviewers
International Journal of Environmental Research and Public Health
RESPONSES TO THE COMMENTS – SECOND ROUND
Title: “Sleep disturbances and health consequences induced by the specificity of nurses' work”
We want to express our great appreciation to You and the reviewers for taking the time and effort necessary to review our manuscript entitled: Sleep disturbances and health consequences induced by the specificity of nurses' work
We carefully considered your comments. They helped us a lot; we appreciate Your patience and willingness to help us to make this manuscript better. Herein, we explain how we revised the paper for a second time based on Your comments and recommendations. All changes are listed in the file below. We have accepted all of your suggestions. Round 2 fixes have been shown in green.
Yours sincerely
Sabina Krupa
Reviewer 1 – Second Round
|
COMMENTS FROM THE EDITOR |
CHANGES MADE |
|
some statements are not in English
" Możliwość poprawy bieżąco 544 , aktywna aktywność fizyczna. W swojej pracy echukwu al., ewidencjonowanie, że również synchronizuje 545 programu ćwiczeÅ„ aerobowych o możliwoÅ›ci poprawiania siÄ™ może poprawiać , poprawia siÄ™ 546 . , że ... 547, jak twierdzisz , że niewystarczajÄ…cy czas badaÅ„ wzajemnych , wtórnych , zwiÄ…zanych z naukÄ… [ 24 ] . 550 pozytywnie promować pielÄ™gniarkÄ™ i sÅ‚użącÄ… do poprawy jakoÅ›ci pracy 551 [25]. [ 26 ] . 554 |
Thank you for your attention. The language has been corrected:
Physical activity can also help to improve sleep. In the article Okechukwu et al. Emphasize that following a moderate-intensity aerobic exercise program can improve both the quality and duration of sleep [23]. In turn, studies by Querstret et al. showed that longer shifts, including nights, and insufficient recovery time between shifts were associated with worse sleep, increased sleepiness and increased fatigue [24]. According to many studies, sleep interventions should be developed that can positively promote the health of nurses and promote effective work performance [25]. In addition, hospital decision makers should consider developing new guidelines to minimize the negative effects of night rotation on the mental health and quality of life of nurses [26]. |

Reviewer 2 Report
Thank you very much for the effort put into the preparation of the revisions. I believe that the quality of the manuscript improved. The minor comments were addresses in the manuscript and I am satisfied with the corrections. None the less, my major comment regarding the conclusions not matching the results is this unsolved.
Line 564 - The Authors still did not link directly the outweigh of the night shift work (the number of the night shifts) with the analyzed symptoms, thus the conclusions are too far going. The conclusions were made upon secondary relations between the number of the night shifts and the insomnia risk, and then between the insomnia risk and mentioned symptoms. What is even more interesting, in the univariate logistic regression the only significant OR was presented for 7-8 shifts/month and it was 0.17, so it does not support the conclusions presented by the Authors (lines 565-566). Line 560 - For more than 8 shifts/month this correlation was not significant.
If the Authors do not provide the analysis supporting these conclusions, please do not present these conclusions as they could potentially mislead the reader.
Lines 544-554 – The Authors typed the text in a foreign language. Please use English in the whole manuscript.
Author Response
Dear August 2022
Editor and Reviewers
International Journal of Environmental Research and Public Health
RESPONSES TO THE COMMENTS – SECOND ROUND
Title: “Sleep disturbances and health consequences induced by the specificity of nurses' work”
We want to express our great appreciation to You and the reviewers for taking the time and effort necessary to review our manuscript entitled: Sleep disturbances and health consequences induced by the specificity of nurses' work
We carefully considered your comments. They helped us a lot; we appreciate Your patience and willingness to help us to make this manuscript better. Herein, we explain how we revised the paper for a second time based on Your comments and recommendations. All changes are listed in the file below. We have accepted all of your suggestions. Round 2 fixes have been shown in green.
Yours sincerely
Sabina Krupa
Reviewer 2 – Second Round
|
COMMENTS FROM THE EDITOR |
CHANGES MADE |
|
Line 564 - The Authors still did not link directly the outweigh of the night shift work (the number of the night shifts) with the analyzed symptoms, thus the conclusions are too far going. The conclusions were made upon secondary relations between the number of the night shifts and the insomnia risk, and then between the insomnia risk and mentioned symptoms. What is even more interesting, in the univariate logistic regression the only significant OR was presented for 7-8 shifts/month and it was 0.17, so it does not support the conclusions presented by the Authors (lines 565-566). Line 560 - For more than 8 shifts/month this correlation was not significant. |
The authors did not mean the number of night shifts per month. It is about the general specificity of a nurse's shift work. We added informations in green:
The specificity of shift work has negative health consequences (especially for sleep problems) in the tested group of nurses.
We also added text in discussion. |
|
Lines 544-554 – The Authors typed the text in a foreign language. Please use English in the whole manuscript. |
Thank you for your attention. The language has been corrected |

Round 3
Reviewer 2 Report
Thank you very much for the effort put in the manuscript revision. I believe that the quality of the article improved. Even though I still have some concerns about the conclusions brought on the basis of the presented results, the Authors' comments are acceptable now.
Please add a separate paragraph summarizing the limitations of the study.
Please revise the manuscript in the context of numerous typos.
Author Response
Dear Reviewer,
thank you for your comments. Below are answers:
- Authors added a separate paragraph summarizing the limitations of the study.
6. Study Limitations
The study was conducted on a group of nurses working night shifts. The study may have limited the various disease entities that may have affected the study group. Besides, the greater number of night shifts could have had an impact on the answers given. Research should also be carried out on other professional groups and the results should be compared also on the basis of research related to possible accompanying disease entities in this group.
- Authors are corrected typos
Kind regards,
Authors